# Paternally expressed imprinted genes establish postzygotic hybridization barriers in *Arabidopsis thaliana*

**Philip Wolff[1,2], Hua Jiang[2], Guifeng Wang[2], Juan Santos-González[2], Claudia Köhler[2]\***

[1]Department of Biology and Zurich-Basel Plant Science Center, Swiss Federal Institute of Technology, Zurich, Switzerland; [2]Department of Plant Biology, Uppsala BioCenter, Swedish University of Agricultural Sciences, Uppsala, Sweden

**Abstract** Genomic imprinting is an epigenetic phenomenon causing parent-of-origin specific differential expression of maternally and paternally inherited alleles. While many imprinted genes have been identified in plants, the functional roles of most of them are unknown. In this study, we systematically examine the functional requirement of paternally expressed imprinted genes (*PEGs*) during seed development in *Arabidopsis thaliana*. While none of the 15 analyzed *peg* mutants has qualitative or quantitative abnormalities of seed development, we identify three *PEGs* that establish postzygotic hybridization barriers in the endosperm, revealing that *PEGs* have a major role as speciation genes in plants. Our work reveals that a subset of *PEGs* maintains functional roles in the inbreeding plant *Arabidopsis* that become evident upon deregulated expression.

## Introduction

Genomic imprinting is an epigenetic phenomenon occurring in mammals and flowering plants that leads to parent-of-origin specific differential expression of maternally and paternally inherited alleles (*Gehring, 2013*). Recent screening of the seed transcriptome in various plant species revealed dozens to several hundreds of novel candidate imprinted genes in maize, rice, castor bean, and *Arabidopsis thaliana* (*Gehring et al., 2011*; *Hsieh et al., 2011*; *Luo et al., 2011*; *Waters et al., 2011*; *Wolff et al., 2011*; *Pignatta et al., 2014*; *Xu et al., 2014*). While few reports demonstrate genes to be temporally imprinted in the plant embryo (*Jahnke and Scholten, 2009*; *Raissig et al., 2013*), the vast majority of imprinted genes has been observed in the endosperm, the ephemeral triploid tissue derived after fertilization of the diploid central cell with a haploid sperm cell. In most angiosperms the endosperm initially develops as a syncytium and cellularizes after a defined number of mitotic divisions (*Li and Berger, 2012*). The right timing of endosperm cellularization is crucial for proper seed development, its failure results in deficient nutrient supply, which causes embryo arrest and eventually seed abortion (*Hehenberger et al., 2012*). In *Arabidopsis*, endosperm cellularization is regulated by, among others, the type I MADS-box transcription factor AGL62. Loss of *AGL62* leads to precocious endosperm cellularization (*Kang et al., 2008*), whereas increased *AGL62* expression correlates with delayed or failed cellularization (*Erilova et al., 2009*; *Tiwari et al., 2010*). Similar effects on endosperm development have been observed in response to interploidy hybridizations. While maternal excess hybridizations cause precocious endosperm cellularization and reduced seed size, the reciprocal cross leads to endosperm cellularization failure and seed abortion in an accession-dependent frequency (*Scott et al., 1998*; *Dilkes et al., 2008*). This phenomenon establishes a postzygotic reproductive barrier by preventing the formation of viable triploid seeds and has been termed 'triploid block' (*Marks, 1966*). Dosage-sensitivity of the endosperm has been proposed to be a consequence of

**\*For correspondence:**
claudia.kohler@slu.se

**Competing interests:** The authors declare that no competing interests exist.

**eLife digest** When plants and animals reproduce sexually, their offspring inherit two copies of every gene, one from each parent, which are arranged in two sets of structures called chromosomes. In some tissues, one gene copy may be switched off—through a process called 'genomic imprinting'—while the other copy remains active. In plants, genomic imprinting is vital for seeds to develop normally. It is particularly important in the tissue that provides nutrients for the growing embryo (the endosperm), in which one of the copies of many genes are switched off. Genes inherited from the male parent that have been imprinted are known as paternally expressed imprinted genes (*PEGs*).

Unlike most animals, it is common for plants to have more than two sets of chromosomes. When plants with different numbers of chromosome sets cross-fertilize each other, their offspring may have three copies of every gene instead of two. These 'triploid' seeds often die because their endosperm fails to develop normally. This is due to the increased activity of imprinted genes, which causes changes in the activity of many other genes in the endosperm. Although it is known that genomic imprinting in the endosperm helps to establish this reproductive barrier, it is not clear what specific roles many of the imprinted genes play.

Here, Wolff et al. switched off several different *PEGs* in the plant *Arabidopsis* to investigate how they affect seed development. The experiments show that in seeds that have the normal two copies of every gene, inactivating these imprinted genes does not affect seed development. However, in triploid seeds, inactivating three of the imprinted genes rescues seeds that would normally die. These genes encode proteins that activate pathways in the endosperm that promote the formation of cell walls, which is a crucial stage in seed development.

Wolff et al.'s findings reveal how imprinted genes in the endosperm establish a barrier to reproduction by preventing seeds produced from crosses between plants with different numbers of chromosome sets from being able to survive. Reproductive barriers are a major obstacle in plant breeding, so understanding how these barriers form may open new avenues for developing new plant varieties.

deregulated imprinted genes that are responsible for interploidy hybridization failure (*Haig and Westoby, 1989*; *Gutierrez-Marcos et al., 2003*; *Kinoshita, 2007*). Indeed, in response to interploidy hybridizations many imprinted genes are deregulated (*Jullien and Berger, 2010*; *Tiwari et al., 2010*; *Wolff et al., 2011*) and the paternally expressed imprinted gene ADMETOS (*ADM*) has been identified to be a causative gene responsible for abortion of triploid seeds upon paternal excess hybridizations in *Arabidopsis* (*Kradolfer et al., 2013*). While the identification of *ADM* provided first evidence that imprinted genes can establish reproductive barriers, the question whether this is a more general phenomenon applying to other imprinted genes as well, remained unresolved.

In this study we investigated the functional role of 15 *PEGs* during seed development in *Arabidopsis*. None of the analyzed *peg* mutants caused qualitative or quantitative abnormalities of diploid seed development, revealing that many *PEGs* do either not have an important functional role in *Arabidopsis* seeds or act redundantly with non-imprinted genes. However, 3 out of ten tested *peg* mutants rescued triploid seed abortion, uncovering a major role of *PEGs* in establishing postzygotic interploidy hybridization barriers.

## Results and discussion

### Impaired *PEG* function does not impact on diploid seed development

Genomic imprinting has been proposed to have a major impact on seed development (*Haig and Westoby, 1989*). We tested this hypothesis by investigating whether loss of *PEG* function would negatively impact on seed development and viability. We examined 15 *PEGs* that were shown to be imprinted at 4 days after pollination (DAP) in reciprocal crosses between Col and Bur-0 accessions (*Wolff et al., 2011*). While At2g36560 (PEG5), At4g05470 (PEG8), At1g67830 (FXG1), At1g17770 (SUVH7), At1g57800 (VIM5) and AT1g48910 (YUC10) were also identified to be imprinted in Col and Ler accessions, At1g11810 (PEG1), At1g49290 (PEG2), At1g60400 (PEG3), At1g66630 (PEG4),

*At3g49770* (*PEG6*), *At3g50720* (*PEG7*), *At5g15140* (*PEG9*), *At1g34650* (*HDG10*) and *At4g31900* (*PKR2*) were not identified as being imprinted in Col and Ler accessions by other studies (*Gehring et al., 2011*; *Hsieh et al., 2011*; *Pignatta et al., 2014*). Due to the lack of small nucleotide polymorphisms (SNPs) for *PEG1, PEG9,* and *PKR2* we only tested the imprinting status of the 6 remaining *PEGs* in reciprocal crosses of Col and Ler accessions at 4 DAP. Parent-of-origin specific expression could be detected for all the genes tested (*Figure 1—figure supplement 1*), revealing that all tested *PEGs* are consistently imprinted in different accessions.

To analyze the functional role of *PEGs* during seed development, we obtained T-DNA insertion mutants for all genes (*Figure 1—figure supplement 2*). We tested the mRNA levels in all mutants that were not yet previously investigated and found them strongly reduced compared to wild type (*Figure 1—figure supplement 3*). Nevertheless, none of the analyzed mutants had increased levels of unfertilized ovules or seed abortion compared to wild-type plants (*Figure 1A*) and neither transmission through the male gametophyte was significantly impaired (Chi–Square test, 1:1 segregation hypothesis; p > 0.3) (*Figure 1—figure supplement 4*). We investigated the possibility that *PEGs* have a positive impact on seed size by pollinating wild-type plants with pollen from heterozygous *peg* mutants and analyzed the size of mature seeds. This analysis revealed that there was no significant (*F*-test; α = 0.01) difference between wild-type and *peg/+* mutant seeds (*Figure 1B*). We furthermore tested the possibility whether PEG function could have a more prominent effect when fitness of the maternal parent was compromised. We completely removed rosette leaves of Col mother plants 2 days prior to emasculation and repeated the crosses with *peg/+* mutants. In agreement with a previous report (*Akiyama and Agren, 2012*), substantial loss of source tissue caused a reduction of seed number in the majority of crosses (*Figure 1—figure supplement 5*). Nevertheless, there was no significant (*F*-test; α = 0.01) difference in seed size between wild-type and *peg/+* mutant seeds (*Figure 1—figure supplement 6*), revealing that there is no general, non-redundant role of *PEGs* in controlling seed development and seed size in *Arabidopsis*.

## *PEGs* establish postzygotic hybridization barriers

We tested the hypothesis that in addition to *ADM* other *PEGs* are involved in building the triploid block. We randomly selected 10 *PEGs* that were, with the exception of *VIM5*, strongly up-regulated in triploid seeds (*Figure 2—figure supplement 1*), and generated double mutants with the *omission of second division1* (*osd1*) mutant. The *osd1* mutation causes the formation of unreduced (2n) male gametes at a frequency of almost 100% (*d'Erfurth et al., 2009*). Therefore, using *osd1* as pollen donor leads to the formation of almost 100% triploid seeds. We used pollen from plants homozygous for both mutations for crosses with wild-type Col mothers. Strikingly, 3 out of ten *peg* mutants were able to rescue triploid seed abortion; while crosses of Col x *osd1* gave rise to 8% non-collapsed seeds, crosses of Col with *suvh7 osd1*, *peg2 osd1* and *peg9 osd1* increased levels of non-collapsed seeds to 53%, 86% and 54%, respectively (*Figure 2A*). The majority of non-collapsed triploid seeds germinated (*Figure 2A*) and developed into viable seedlings (*Figure 2B*). Independent mutant alleles for *suvh7* and *peg9* introduced into the *osd1* background caused a similar rescue effect on triploid seeds (*Figure 2—figure supplement 2*). As for *peg2* no second mutant allele could be identified, we complemented the mutant phenotype with a genomic *PEG2::PEG2* construct that restored triploid seed inviability, revealing that the *peg2* phenotype is indeed caused by failure of PEG2 function (*Figure 2—figure supplement 2*). To test whether maternal loss of PEG function would impact on triploid seed rescue, we pollinated either *osd1* or *peg osd1* mutant pollen onto maternal plants mutant for the corresponding *peg*. However, no maternal effect on triploid seed rescue could be detected (*Figure 2C*), revealing no impact of the maternal alleles of *SUVH7*, *PEG2* and *PEG9* on the triploid block. Consistently, *SUVH7* and *PEG2* remained imprinted in triploid seeds (*Figure 2D*). *PEG9* has no polymorphism between Col and Ler accessions and could not be tested.

## Rescue of triploid seed development by *peg* mutants occurs by different mechanisms

*SUVH7* encodes for a putative histone-lysine N-methyltransferase that however did not have histone methyltransferase activity against H3K9 in *in vitro* assays (*Ebbs and Bender, 2006*). *PEG2* encodes for an unknown protein with no predicted structural domains, while *PEG9* encodes for a galactose mutarotase, which catalyzes the first step of the Leloir pathway, the conversion of beta-D-galactose to

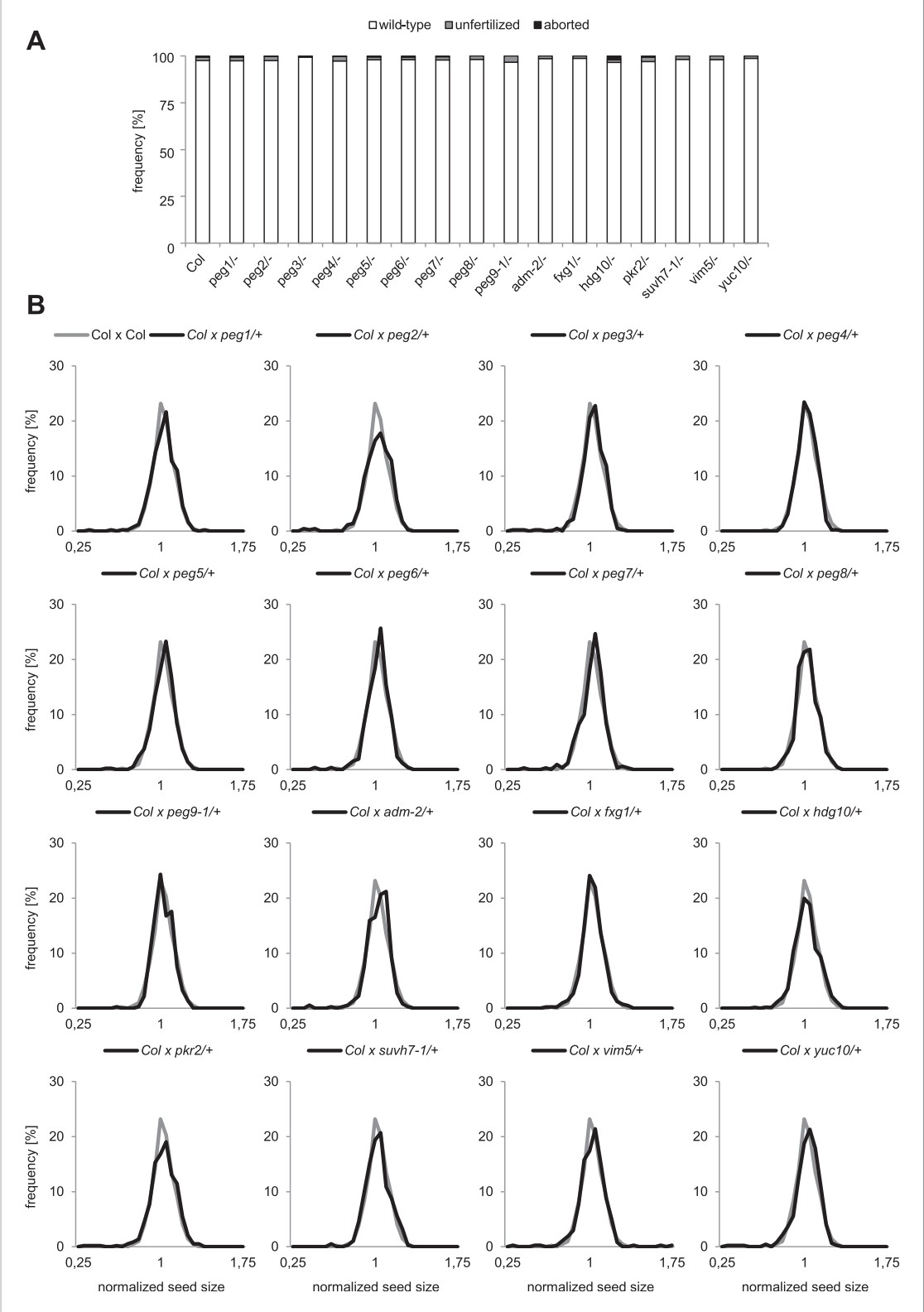

**Figure 1**. Impact of *PEG* function on diploid seed development. (**A**) Percentage of normal, unfertilized, and aborted seeds from self-fertilized wild-type and homozygous *peg* mutant plants. A minimum of 300 seeds was analyzed for each genotype. (**B**) Size measurements of mature seeds derived from crosses of maternal Col plants pollinated with heterozygous *peg* pollen (black line) were normalized, plotted on a histogram and distribution was compared with wild type control crosses (grey line). A minimum of 400 seeds was analyzed for each cross.

*Figure 1. continued on next page*

*Figure 1. Continued*

The following figure supplements are available for figure 1:

**Figure supplement 1**. Imprinting of *PEGs* in Col/L*er*.

**Figure supplement 2**. T-DNA insertions in *peg* mutants.

**Figure supplement 3**. *PEG* expression in *peg* mutants.

**Figure supplement 4**. Transmission analysis of *peg* mutant alleles through the male germ line.

**Figure supplement 5**. Effect of stress on seed set in Col x *peg/+* crosses.

**Figure supplement 6**. Seed size analysis under stress conditions.

alpha-D-galactose (*Holden et al., 2003*). Galactose is a major constituent of the *Arabidopsis* endosperm cell wall in the form of pectic homogalacturan (*Lee et al., 2012*) and altered *PEG9* activity could impact on cell wall composition. We tested whether rescue of triploid seed viability by *suvh7*, *peg2* and *peg9* was associated with restored endosperm cellularization. While the endosperm of triploid seeds was completely uncellularized at 8 DAP, endosperm cellularization in triploid *suvh7*, *peg2* and *peg9* seeds was almost complete at 8 DAP; only the over-proliferated chalazal cyst remained uncellularized (*Figure 2E*). Failure of endosperm cellularization is responsible for embryo arrest (*Hehenberger et al., 2012*). Consistently, embryos of *peg2*, *peg9*, and *suvh7* triploid seeds developed, albeit delayed compared to wild-type diploid embryos (*Figure 2E*).

Triploid seed rescue by *adm* is associated with strongly decreased mRNA levels of type I *AGAMOUS-LIKE* MADS-box genes (*AGLs*) as well as *PEGs* (*Kradolfer et al., 2013*). To test whether triploid rescue is generally connected with decreased expression of *AGLs* and *PEGs*, we generated whole-genome transcriptome data of seeds from triploid *adm*, *suvh7* and *peg2* mutants and analyzed expression of *AGLs* and *PEGs* that had increased mRNA levels in triploid seeds. Triploid seed rescue by *adm* and *suvh7* was associated with a similar decrease of mRNA levels of many *AGLs* and *PEGs* (*Figure 3A,B*), suggesting that ADM and SUVH7 share a common mode of action. Strikingly, while *peg2* had the strongest effect on triploid seed rescue (*Figure 2A*), the effect on gene expression was weakest among the three mutants. In particular, *AGLs* and *PEGs* that were strongly affected in *adm* and *suvh7* triploid seeds were only weakly affected in *peg2* triploid seeds (*Figure 3A,B*). This suggests that *peg2*-mediated triploid seed rescue occurs independently of normalized *AGL* and *PEG* expression and may affect a pathway downstream of either *AGLs* or *PEGs*. In support of this view, most genes down-regulated in triploid *peg2* seeds were similarly down-regulated in triploid *adm* and *suvh7* seeds (*Figure 3C*, *Figure 3—source data 1*), suggesting that all three mutants affect a common downstream pathway. Genes commonly down-regulated in all three mutants were significantly enriched for genes involved in carbohydrate metabolism and in particular genes encoding for polygalacturonases (*Figure 3—figure supplement 1*). Polygalacturonan is the backbone of the major primary cell wall component pectin, which is degraded by polygalacturonases (*Atmodjo et al., 2013*). Pectin degradation is assumed to be a key step in the deconstruction of plant cell walls (*Xiao et al., 2014*), therefore suppression of pectin hydrolysis in *adm*, *suvh7*, *peg2*, and *peg9* may be the key mechanism to induce endosperm cellularization and restore triploid seed viability. In agreement with this notion, *PEG9* encodes an enzyme that acts on galactose, the carbohydrate building pectin.

## Endosperm cellularization failure correlates with deregulated expression of genes encoding for pectin degrading enzymes

To test the hypothesis that endosperm cellularization failure in triploid seeds is a consequence of disturbed pectin degradation pathways, we analyzed expression of 173 genes acting in pectin degrading pathways (pectate lyases (GO:0030570), pectin methylesterases (GO:0030599) and polygalacturonases (GO:0004650)). Of those, 33 genes were upregulated in triploid seeds (*Figure 4A*). In contrast, expression of pectin biosynthesis genes (*Atmodjo et al., 2013*) was not

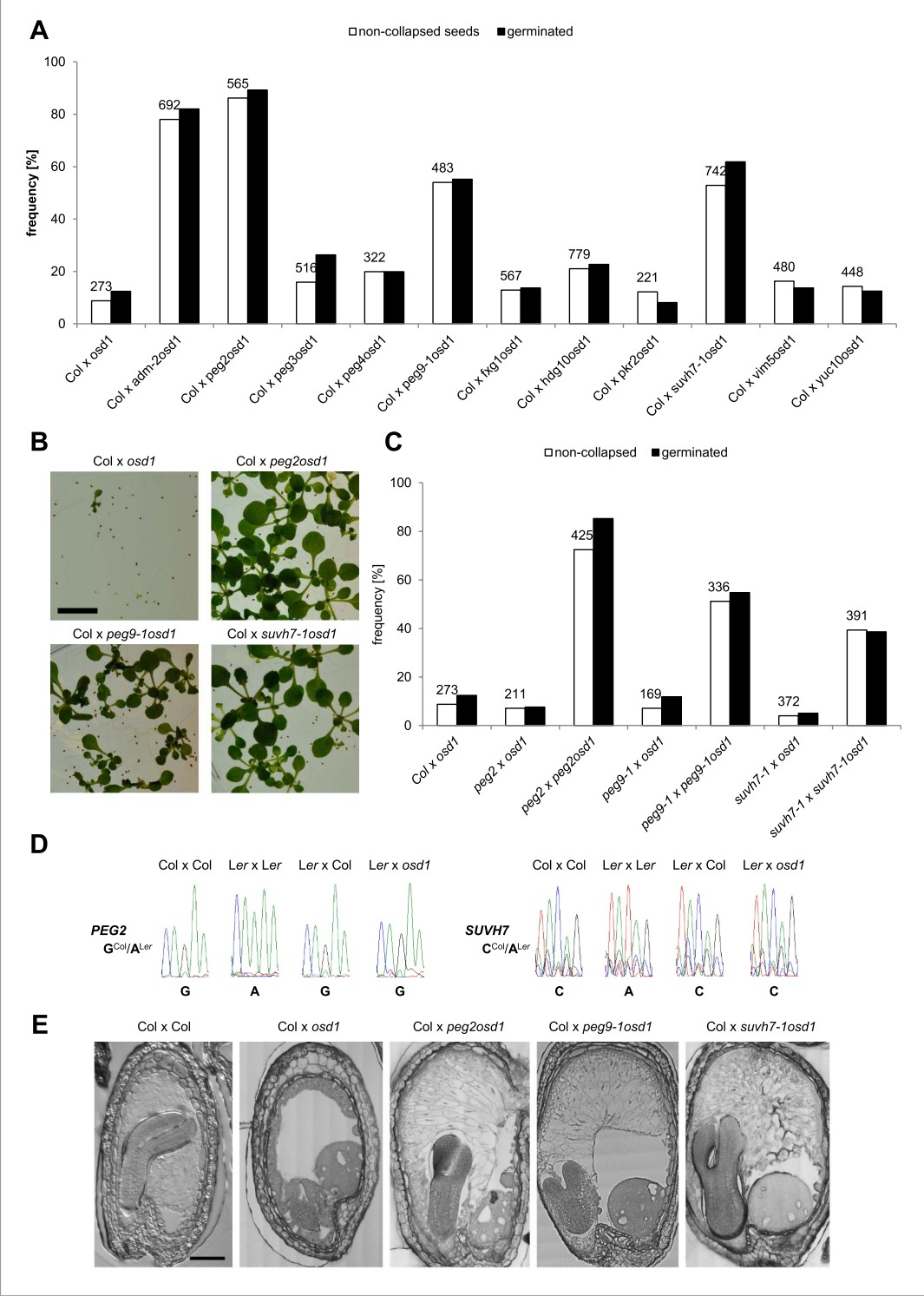

**Figure 2**. *PEGs* establish interploidy hybridization barriers. (**A**) Percentages of non-collapsed and germinated seeds of wild-type plants pollinated with *osd1* and *peg osd1* pollen. Numbers on top of bars correspond to number of analyzed seeds. (**B**) Triploid seedlings 14 days after germination. Scale bar, 1 cm. (**C**) Percentages of non-collapsed and germinated seeds of *peg* mutant plants pollinated with *osd1* and *peg osd1* pollen. Numbers on top of bars correspond to number of analyzed seeds. (**D**) *PEG2* and *SUVH7* remain imprinted in diploid and triploid seeds. Siliques of crosses of L*er* plants pollinated with Col or *osd1* pollen were harvested at 4 DAP and imprinted expression was tested by PCR and subsequent DNA sequencing. Siliques of Col and L*er* plants pollinated with Col

*Figure 2. continued on next page*

*Figure 2. Continued*

and L*er* pollen were used as controls. (**E**) Sections of seeds derived from crosses of Col plants pollinated with Col, *osd1*, *peg2 osd1*, *peg9-1 osd1* and *suvh7-1 osd1* pollen at 8 DAP. Scale bar, 0.1 mm.
The following figure supplements are available for figure 2:

**Figure supplement 1**. *PEG* expression in triploid seeds.

**Figure supplement 2**. Analysis of independent mutant alleles for *suvh7* and *peg9* and genomic complementation of *peg2*.

---

negatively affected in triploid seeds (*Figure 4B*), suggesting that pectin degradation rather than pectin biosynthesis is disturbed in triploid seeds. Most of the 33 pectin degradation genes remained repressed until the heart stage of embryo development in wild-type seeds (*Figure 4C*) and became expressed at the cotyledon stage of embryo development, concomitantly with endosperm degradation (*Figure 4C*). In concordance with restored endosperm cellularization in triploid *peg* mutants, expression of 27 out of 33 pectin degradation genes that were upregulated in triploid seeds was normalized in at least one of the *peg* mutants (*Figure 4D*). Pectins are polymerized and methylesterified in the golgi and secreted into the cell wall as highly methylesterified forms. Subsequently, they can be modified by pectinases such as pectin methylesterases that catalyse the demethylesterification of homogalacturonans releasing acidic pectins, which can be visualized by ruthenium red binding (*Downie et al., 1998*; *Micheli, 2001*). Before endosperm cellularization, the ruthenium red signal was similar between diploid and triploid seeds (*Figure 4E*; *Figure 4—figure supplement 1*), suggesting that there were no major differences in pectin synthesis and degradation between diploid and triploid seeds. At 6 DAP the wild-type endosperm was largely cellularized and only a weak ruthenium red signal could be detected (*Figure 4E*; *Figure 4—figure supplement 1*), in agreement with pectin being deposited in the cell wall in a highly methylesterified form that is less intensively stained by ruthenium red (*Micheli, 2001*). Consistent with the expression of genes encoding for pectin degrading enzymes in 6 DAP triploid seeds (*Figure 4D*), the ruthenium red signal in the uncellularized endosperm at 6 DAP seeds was substantially weaker compared to the signal in 4DAP seeds (*Figure 4E*; *Figure 4—figure supplement 1*).

Together, our data reveal that a subset of *PEGs* has a functional role in establishing interploidy hybridization barriers and that bypass of the barriers occurs by mechanisms converging on endosperm cellularization likely by suppression of pectin degradation. Thus, a subset of *PEGs* maintains a functional role in *Arabidopsis*, which is only revealed upon deregulation in triploid seeds.

## Material and methods

### Plant material and growth conditions

*A. thaliana* mutants *adm-2* (*Kradolfer et al., 2013*), *pkr2-1* (*Aichinger et al., 2009*) and *yuc10* (*Cheng et al., 2007*) have been described previously. *peg1* (SAIL_659_B07), *peg2* (SALK_143382), *peg3* (SALK_000257), *peg4* (GABI_022C09), *peg5* (SALK_142234), *peg6* (WiscDsLox428C06), *peg7* (SALK_103601), *peg8* (SALK_022399), *peg9-1* (SALK_102165), *peg9-2* (SALK_047888), *fxg1* (SAIL_888B03), *hdg10* (SALK_116071), *suvh7-1* (GABI_037C06), *suvh7-2* (SALK_112939), *suvh7-3* (WiscDSLox297300_14H) and *vim5* (SALK_033892) T-DNA insertion mutants were received from the *Arabidopsis* stock center (arabidopsis.info) and mutant plants were identified using primers listed in *Supplementary file 1*. The *osd1* mutant (*d'Erfurth et al., 2009*) was kindly provided by Raphael Mercier. Being originally identified in the Nossen background, the mutant was introgressed into Col by repeated backcrossing over five generations. Plants were grown in a growth cabinet under long day photoperiods (16 hr light and 8 hr dark) at 22°C. After 10 days, seedlings were transferred to soil and plants were grown in a growth chamber at 60% humidity and daily cycles of 16 hr light at 22°C and 8 hr darkness at 18°C. For all crosses, designated female plants were emasculated and the pistils were hand-pollinated 2 days after emasculation.

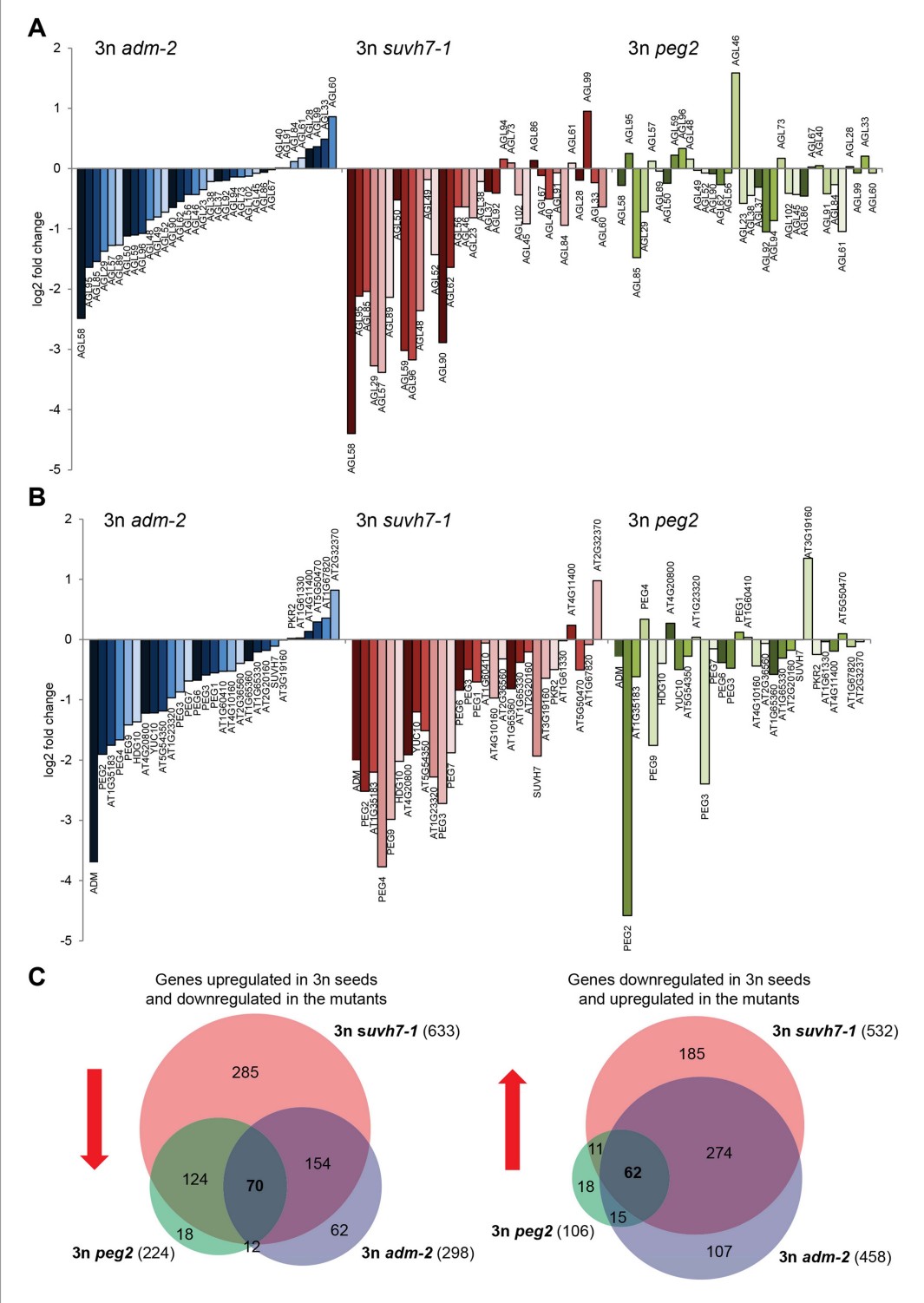

**Figure 3**. Transcriptome analysis of *AGL* genes and *PEGs* in triploid *adm-2*, *suvh7-1* and *peg2* seeds. (**A**) Log2 fold change expression of *AGLs* in triploid *adm-2*, *suvh7-1* and *peg2* mutants compared to triploid wild-type seeds. Only *AGLs* were tested that were up-regulated in triploid wild-type seeds. (**B**) Log2 fold change expression of *PEGs* in triploid *adm-2*, *suvh7-1* and *peg2* mutants compared to triploid wild-type seeds. Only *PEGs* were tested that were up-regulated in triploid wild-type seeds. (**C**) Left panel: Venn diagram showing overlap of genes being up-regulated in seeds derived from wild type x *osd1* crosses (signal log ratio [SLR] > 1, p < 0.05) and down-regulated in wild type x *adm-2 osd1* (SLR < −1, p < 0.05), wild type x *suvh7-1 osd1* (SLR < −1, p < 0.05), and wild type x *peg2 osd1* (SLR < −1,

*Figure 3. continued on next page*

*Figure 3. Continued*

p < 0.05) Hypergeometric testing was used to test for significance of overlap; p = 5.853e-09. Right panel: Venn diagram showing overlap of genes being down-regulated in seeds derived from wild type x *osd1* crosses (SLR < −1, p < 0.05) and up-regulated in wild type x *adm-2 osd1* (SLR >1, p < 0.05), wild type x *suvh7-1 osd1* (SLR >1, p < 0.05), and wild type x *peg2 osd1* (SLR >1, p < 0.05). Hypergeometric testing was used to test for significance of overlap; p = 4.622 e−16.

The following source data and figure supplement are available for figure 3:

**Source data 1**. List of genes deregulated in seeds derived from interploidy crosses.

**Figure supplement 1**. Gene ontology analysis.

## RNA sequencing and expression analysis

For RNA sequencing, seeds from 20 siliques of L*er* x Col, L*er* x *osd1*, L*er* x *adm-2 osd1*, L*er* x *suvh7-1 osd1*, and L*er* x *peg2 osd1* were harvested at 6 DAP into RNA later (Sigma–Aldrich, St Louis, Missouri) in duplicates and homogenized (Silamat S5; IvoclarVivadent, Germany) using glass beads (1.25–1.55 mm; Carl Roth, Germany). RNA was extracted following a modified protocol for the RNAqueous kit (Ambion, Life Technologies, Carlsbad, California). RNA was purified by Qiagen RNeasy Plant Mini Kit (Qiagen, Germany) after residual DNA removed by 2 µL DNaseI (Thermo-Scientific, Waltham, Massachusetts). Libraries were prepared using the Truseq RNA Sample Preparation Kit (Illumina, San Diego, California) and sequenced at the SciLife Laboratory (Uppsala, Sweden) on an Illumina HiSeq2000 on two lanes in 100-bp paired-end mode. Sequencing reads have been deposited as fastq files in the Gene Expression Omnibus (*Santos-González, 2015*). For qPCR expression analysis three siliques of each cross were harvested, flash-frozen in liquid nitrogen and samples were disrupted using glass beads (1.25–1.55 mm; Carl Roth) and a Silamat S5 machine (IvoclarVivadent). RNA was extracted using the RNeasy Plant Mini Kit (Qiagen) according to the manufacturer's instructions and residual DNA was removed using the RNase-free DNase set (Qiagen). cDNA was synthesized using the first-strand cDNA synthesis kit (Thermo-Scientific) according to the manufacturer's instructions. Quantitative real-time PCR was performed using an iQ5 Real-Time PCR detection system (Bio-Rad, Hercules, California) in triplicates using Maxima SYBR green master mix (Thermo-Scientific). Results were analyzed as described by *Simon (2003)* using *ACTIN11* as a reference gene and qPCR primers are listed in *Supplementary file 1*.

## High-throughput RNA sequence analysis

Gene expression data were quality trimmed and mapped to the *Arabidopsis* TAIR10 reference genome using TopHat v2.0.10 (*Trapnell et al., 2009*). A maximum of 1 alignment to the reference was allowed for any given read, and the minimum anchor length was 10 bases. Differentially regulated genes across the two replicates were detected using the rank product method (*Breitling et al., 2004*) as implemented in the Bioconductor RankProd Package (*Hong et al., 2011*). The test was run with 100 permutations and gene selection was corrected for multiple comparison errors using a pfp (percentage of false prediction) < 0.05. Gene ontology (GO) categories were identified using AtCOECiS (*Vandepoele et al., 2009*).

## Imprinting analysis

To determine the imprinting status of selected genes, RNA was extracted from crosses Col x Col, L*er* x L*er*, L*er* x Col and L*er* x *osd1* at 4 DAP. Primers used for allele specific expression analysis are specified in *Supplementary file 1* and PCR products were analyzed by DNA sequencing.

## Microscopy

Seeds were fixed and embedded with Technovit 7100 (Heraeus, Germany) as described (*Hehenberger et al., 2012*). Five-micrometer sections were prepared with an HM 355 S microtome (Microm, Germany) using glass knives. Sections were stained for 1 min with 0.1% toluidine blue and washed three times with distilled water. For pectin analysis, sections were stained for 45 min with 0.025%

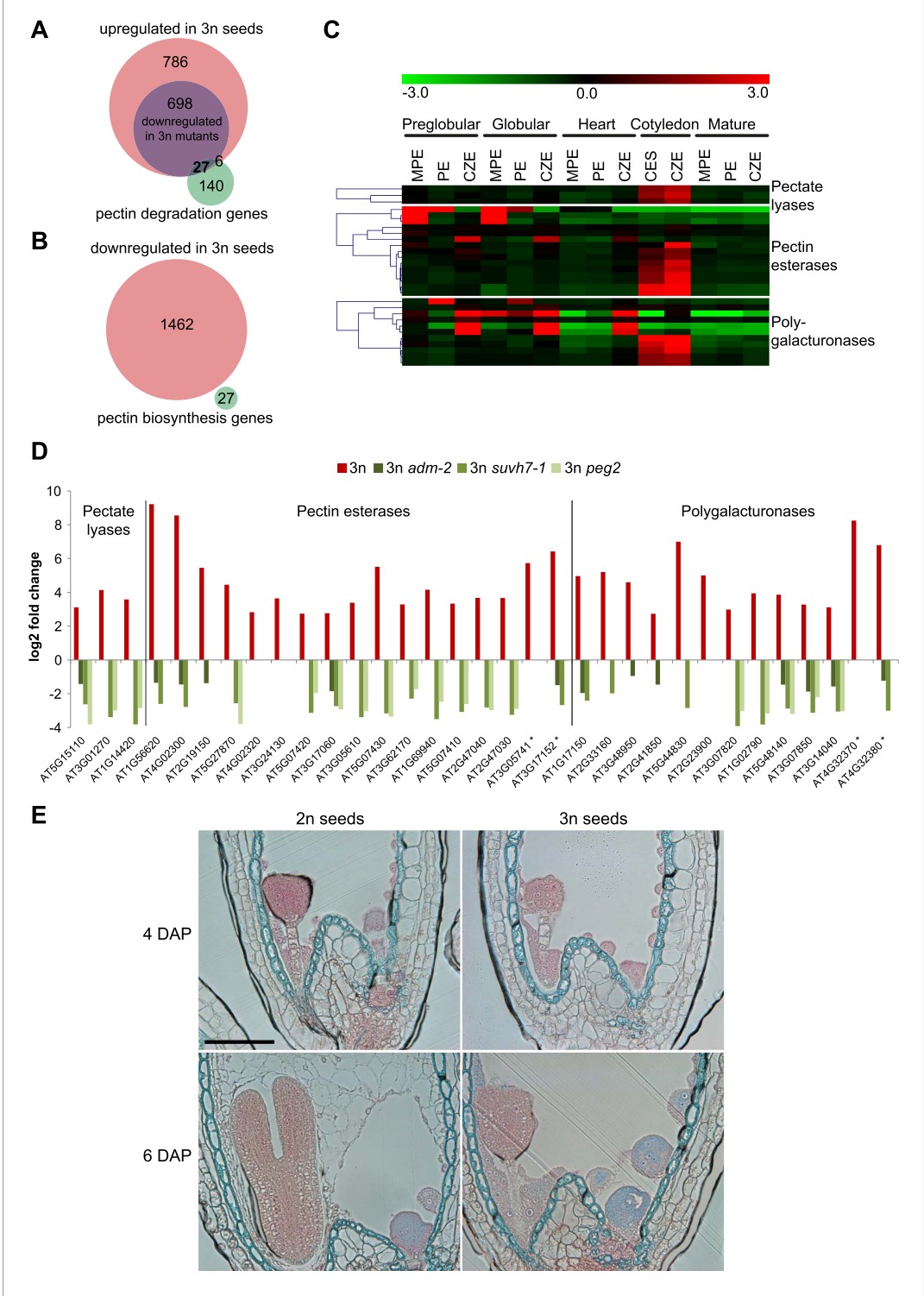

**Figure 4**. Analysis of pectin biosynthesis and degradation genes in diploid and triploid seeds. (**A**) Venn diagram showing overlap of pectin degradation genes and genes being upregulated in seeds derived from wild type x *osd1* crosses (signal log ratio [SLR] > 1, p < 0.05) and genes being downregulated in either wild type x *adm-2 osd1* (SLR < −1, p < 0.05), wild type x *suvh7-1 osd1* (SLR < −1, p < 0.05), or wild type x *peg2 osd1* (SLR < −1, p < 0.05). Hypergeometric testing was used to test for significance of overlap, p = 4.048 e−10. (**B**) Venn diagram showing overlap of pectin biosynthesis genes and genes being down-regulated in seeds derived from wild type x *osd1* crosses (SLR <1, p < 0.05). (**C**) Cluster analysis of pectin degradation genes that are upregulated in triploid wild-type

*Figure 4. continued on next page*

*Figure 4. Continued*

seeds, based on their expression in endosperm during different stages of diploid seed development (*Belmonte et al., 2013*). Each row represents a gene, and each column represents a tissue type. Tissue types are: micropylar (MPE), peripheral (PE), chalazal (CZE) and cellularized endosperm (CES) derived from seeds containing embryos of the preglobular stage to the mature stage. Red or green indicate tissues in which a particular gene is highly expressed or repressed, respectively. (**D**) Log2 fold change expression of pectin degradation genes in triploid wild-type seeds (compared to diploid wild-type seeds) and triploid *adm-2*, *suvh7-1* and *peg2* mutants (compared to triploid wild-type seeds). Genes marked by an asterisk are not included in the seed transcriptome dataset (*Belmonte et al., 2013*) and are therefore not included in panel (**C**). (**E**) Ruthenium red staining of sections of seeds derived from crosses of Col plants pollinated with Col and *osd1* at 4 DAP and 6 DAP. Red color marks the presence of demethylesterified pectin, blue color is derived from the counterstain with toluidine blue. A minimum of 100 seeds was analyzed for each cross. Scale bar, 0.1 mm.

The following figure supplement is available for figure 4:

**Figure supplement 1**. Additional ruthenium red staining of seeds.

ruthenium red, counterstained for 1 min with 0.1% toluidine blue and washed three times with distilled water. At least ten seeds were analyzed per genotype and timepoint. Microscopy was performed using a DMI 4000B microscope with DIC optics (Leica, Germany). Images were captured using a DFC360 FX camera (Leica) and processed using Photoshop CS5 (Adobe, San Jose, California).

## Seed size analysis

Siliques were harvested when they turned brown and prior dehiscence. Mature seeds were separated from the siliques, spread on a document scanner with backlight unit (Scanmaker i800; Mikrotek, Taiwan) and analyzed as previously described (*Herridge et al., 2011*). Measurements were normalized by dividing by the average seed area and plotted on histograms. To determine statistical differences between sample and control crosses, Bonferroni corrected F-Tests were performed at a level of $\alpha = 0.01$.

## Germination and transmission analysis

Seeds were surface sterilized in a container using chlorine gas (10 ml hydrochloric acid plus 50 ml sodium hypochlorite) and incubated for up to 3 hr. To determine germination frequency, seeds were plated on ½ MS media containing 1% sucrose, stratified at 4°C for 2 days in the dark and grown in a growth cabinet under long day photoperiods (16 hr light and 8 hr dark) at 22°C for 10 days. For transmission analysis, seedlings were harvested after 12 days and genotyped using primers specified in *Supplementary file 1*.

## Generation of plasmids and transgenic lines

For the generation of a Col *PEG2::PEG2* construct, *At1g49290* including 1.5 kb of upstream secuence was amplified by PCR using primers specified in *Supplementary file 1*. The product was cloned into pENTR/D-TOPO (Invitrogen, Carlsbad, California), followed by clonase reaction with the pB7FWG2 vector (*Karimi et al., 2002*), from which the 35S promoter was removed. The Col *PEG2::PEG2* construct was introduced into the *peg2/− osd1/+* double mutant using *Agrobacterium tumefaciens*-mediated transformation (*Clough and Bent, 1998*) and transformants were selected on ½ MS media containing 30 mg/L phosphinotricin. Independent T2 lines were selected for single locus insertions and eight independent *PEG2::PEG2* lines (in *peg2/− osd1/−* background) were used for pollinations onto Col wild-type plants.

## Acknowledgements

We thank Lars Hennig for critical comments on the manuscript. Sequencing was performed by the SNP&SEQ Technology Platform, Science for Life Laboratory at Uppsala University, a national infrastructure supported by the Swedish Research Council (VRRFI) and the Knut and Alice Wallenberg Foundation. This research was supported by a European Research Council Starting Independent Researcher grant (to CK), a grant from the Swedish Science Foundation (to CK) and a grant from the Knut and Alice Wallenberg Foundation (to CK).

## Additional information

### Funding

| Funder | Grant reference | Author |
|---|---|---|
| European Research Council (ERC) | ERC-2011-StG - 280496 | Claudia Köhler |
| Vetenskapsrådet (Swedish Research Council) | 2014-3820 | Claudia Köhler |
| Knut och Alice Wallenbergs Stiftelse (Knut and Alice Wallenberg Foundation) | | Claudia Köhler |
| Stiftelsen Olle Engkvist Byggmästare | | Claudia Köhler |

The funders had no role in study design, data collection and interpretation, or the decision to submit the work for publication.

### Author contributions
PW, Conception and design, Acquisition of data, Analysis and interpretation of data, Drafting or revising the article; HJ, GW, Conception and design, Acquisition of data, Analysis and interpretation of data; JS-G, Analysis and interpretation of data, Drafting or revising the article; CK, Conception and design, Analysis and interpretation of data, Drafting or revising the article

### Author ORCIDs
Philip Wolff, http://orcid.org/0000-0002-8915-553X
Claudia Köhler, http://orcid.org/0000-0002-2619-4857

## Additional files

### Supplementary file
• Supplementary file 1. Primers used in this study.

### Major dataset
The following dataset was generated:

| Author(s) | Year | Dataset title | Dataset ID and/or URL | Database, license, and accessibility information |
|---|---|---|---|---|
| Santos-González J | 2015 | Paternally expressed imprinted genes establish postzygotic hybridization barriers in Arabidopsis thaliana | http://www.ncbi.nlm.nih.gov/geo/query/acc.cgi?acc=GSE62401 | Publicly available at the NCBI Gene Expression Omnibus (GSE62401). |

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
