## [Decision Letter]

[Editors’ note: a previous version of this study was rejected after peer review, but the authors submitted for reconsideration. The previous decision letter after peer review is shown below.]

Thank you for choosing to send your work entitled “Paternally expressed imprinted genes establish postzygotic hybridization barriers in *Arabidopsis thaliana”* for consideration at *eLife*. Your full submission has been evaluated by Detlef Weigel (Senior Editor) and two peer reviewers, and the decision was reached after discussions between the reviewers. Based on our discussions and the individual reviews below, we regret to inform you that your work cannot be considered for publication in *eLife* at this time.

There was agreement that the manuscript reports on an interesting topic, and that it is a nice extension of your recent paper in Developmental Cell. The reviewers were excited about the possibility that *PEGs* converge on pectin degradation and thereby overcome the triploid block. Unfortunately, it is currently unclear where and how the *PEGs* converge to affect pectin degradation genes, whether pectin degradation is indeed altered, and whether the changes in expression of pectin degradation genes are cause or consequence of the suppression of the triploid block. Furthermore, there is only weak support for the claim that, in agreement with parental conflict theory, *PEGs* function as growth regulators. If these *PEGs* functioned as positive regulators of embryo growth during parental conflict, would one not expect *PEG* mutations to have paternal effects on embryo growth (i.e. reduced resources/growth of embryos)?

We would be willing to reconsider as a new submission a manuscript that directly shows altered pectin degradation, and demonstrates that this is the direct cause of triploid block suppression.

Reviewer #1:

The dosage effect of imprinted genes has been associated with postzygotic hybridization barriers in the endosperm. A paternally imprinted gene in *Arabidopsis*, *ADMETOS* (*ADM*), influences endosperm cellularization and contributes to seed abortion in interploidy crosses, a phenomenon known as the triploid block. In this manuscript, the authors show that three additional paternally imprinted genes contribute to the triploid block. They hypothesise that the triploid block is circumvented by the suppression of pectin degradation. The identification of four imprinted genes that contribute to the triploid block in plants gives important insight into this developmental process and it is potentially influential to the field of imprinting.

Although I potentially support the publication of the present research, my main concern is that the effect of the mutations on the three imprinted genes is not fully understood and thus their precise role in triploid block remains unknown.

1) It is unclear why only one mutant allele of *SUVH7*, *PEG2* and *PEG9* was tested in the interploidy analysis. To avoid any misinterpretation of phenotypes, two independent mutant alleles for each gene should be tested.

2) The authors postulate that suppression of pectin hydrolysis by *PEGs* could be the mechanism that restores triploid viability. However, this is highly speculative especially considering that the precise functions of three of the *PEGs* remain known.

3) I do not see how the data support the conclusion that *PEGs* function as growth regulators and that this agrees with the predictions of the parental conflict theory.

Reviewer #2:

In this manuscript, the authors investigated the functional roles of 15 *PEGs* during *Arabidopsis* seed development. Although they did not find obvious functions of these genes in wild-type seeds, clever genetic experiments revealed that at least three *PEGs* are required for the so-called triploid block, a post-zygotic reproductive barrier that originates in the endosperm. These results substantiate their earlier findings that imprinted genes, such as *ADM*, can establish post-zygotic hybridization barriers and thus may be key speciation genes. However, they do not establish a strong connection between their genetic results and expression analyses. If the authors want to make the claim summarized in the second to last sentence in the Abstract, then more work is needed with either existing resources or new experiments. That is, while the pathways appear to converge on genes that decrease pectin degradation, which may increase cellularization and subsequent embryo viability, it is impossible to evaluate how likely this is a cause or correlation based on the presented data. Overall, I agree that this work advances the understanding of the functional roles of imprinted genes in plants, but think more could be presented to get an idea of the underlying molecular mechanism.

At the very least, the authors should give more information relating to the hypothesis that suppression of pectin degradation in the triploid *PEG* mutants contributes to increased viability. This seems like an important aspect of their manuscript, but GO enrichments of commonly down-regulated genes are not necessarily strong indicators of common functions. For instance, they should mention how many genes were commonly down-regulated in the three triploid mutants (Figure 3 shows 70 but this was only for genes up-regulated in triploid seeds). And they should show the levels of the 18 (?) commonly down-regulated genes suggested to be involved in pectin degradation (18 based on Figure 3—figure supplement 1, but perhaps some of these may be overlapping?) in wild-type, triploid and triploid *peg* mutant backgrounds. It would be interesting to see the transcript levels of these pectin degradation genes in wild-type endosperm and these should be available from the microarray datasets generated by the Harada and Goldberg labs (Belmonte (2013) PNAS). Finally, they should stain the triploid seeds/endosperm (and controls) with a pectin dye such as Ruthenium Red to test for decreased pectin levels at multiple time points prior to cellularization. This may address whether decreased pectin is a cause or consequence of increased cellularization in the triploid mutants.

[Editors’ note: what now follows is the decision letter after the authors submitted for further consideration.]

Thank you for submitting your work entitled “Paternally expressed imprinted genes establish postzygotic hybridization barriers in *Arabidopsis thaliana”* for peer review at *eLife*. Your submission has been favorably evaluated by Detlef Weigel (Senior Editor) and two reviewers.

The reviewers have discussed the reviews with one another and the Reviewing editor has drafted this decision to help you prepare a revised submission.

As you will see, both reviewers agree that this manuscript has substantially improved. One of the reviewers asked for Ruthenium Red staining of the 3n *peg* mutants. I had asked you about this, and the reviewers and I have discussed your response. We understand that the reciprocal link between cellularization and PME activity makes it difficult to properly interpret Ruthenium Red staining. Given that this is the case, and assuming you would like to keep the stainings you have, at least the conclusions drawn from the experiment should not be stated as strongly as they are now.

Reviewer #1:

The revised manuscript has improved significantly. The authors clearly show a direct link between three imprinted *PEGs* and interploidy hybridization barriers through the suppression of pectin degradation in endosperm.

This work advances our understanding of imprinting in plants and reveals some of the underlying molecular mechanisms.

Reviewer #2:

Wolff et al. investigated the functional roles of 15 *PEGs* during *Arabidopsis* seed development, and found evidence that three of them contribute to the triploid block, a post-zygotic reproductive barrier that originates in the endosperm. These results are consistent with their earlier findings that imprinted genes, such as *ADM*, can establish post-zygotic hybridization barriers and thus may be key speciation genes. In this manuscript, the authors suggest that *PEGs* can contribute to the triploid block by promoting pectin degradation and thus increasing endosperm cellularization and subsequent seed viability. The use of multiple alleles, or complementation, and more detailed expression analyses builds a solid case for their hypothesis. The Ruthenium Red staining is also consistent with their hypothesis, although it is difficult to interpret staining patterns/intensities between cellularized and uncellularized endosperm. Nevertheless, this manuscript advances the understanding of imprinted gene functions in plants and unveils a mechanism by which this can occur.

The authors should perform Ruthenium Red staining on the 3n *peg* mutants (e.g. 3n *adm-2*, 3n *suv7-1*, 3n *peg2*) to check whether pectin degradation defects are rescued by mutating *PEGs*.

Minor comments:

1) Remove or reword the last sentence of the Results/Discussion. It doesn't read like science text.

2) Explain how seed size (Figure 1) was normalized.

3) Ruthenium Red staining looks less in 3n 6 DAP compared to 4 DAP. Is this representative? Also, it is interesting that there is less staining in the chalazal endosperm. Perhaps it is worth commenting on this.

4) In Figure 1—figure supplement 1, I assume these were for seed/endosperm-specific genes. Otherwise, one would expect a strong maternal bias in all crosses. This should be noted in the text or figure legend.

5) Include error bars in Figure 1—figure supplement 3.

---

## [Author Response]

[Editors’ note: the author responses to the first round of peer review follow.]

We were pleased to see that the reviewers found our work of interest and that it could be improved based on their suggestions. The major concern was the lack of supportive data for the role of pectin degradation in the triploid block response. To address this concern we have included additional data in Figure 4 and Figure 4—figure supplement 1 that show that triploid seeds have reduced levels of methyl-esterified pectin, in agreement with increased expression of enzymes degrading this substrate. We furthermore have included data for additional alleles of those mutants that suppress triploid seed abortion (Figure 2—figure supplement 2).

Our manuscript addresses an open question in the imprinting field concerning the functional role of imprinted genes during plant development. Recent genome-wide efforts discovered dozens to several hundreds of imprinted genes in various plant species; however, our understanding of the functional role of imprinted genes remained sparse. In this study we tested whether paternally expressed imprinted genes (*PEGs*) have a functional role during seed development in *Arabidopsis thaliana*. We report that loss of function of 15 *PEGs* did not impair seed development and neither caused a quantitative effect on seed size. Strikingly however, we identified three *PEGs* that establish postzygotic hybridization barriers, preventing successful hybridizations of plants that differ in ploidy. Therefore, together with the previously identified *PEG ADM,* we have discovered four *PEGs* in *Arabidopsis* that establish barriers to interploidy 2 hybridizations, revealing that PEGs have a major role as speciation genes in plants. Based on genome-wide transcriptome studies of three *peg* mutants we reveal that bypass of the hybridization barrier likely occurs by different mechanisms that converge on suppression of endosperm cell wall formation by suppressing pectin degradation. Collectively, our work reveals that a subset of *PEGs* maintains functional roles in the inbreeding plant *Arabidopsis* that become evident upon deregulated expression. We strongly believe that our results are of general interest to a broad public.

Reviewer #1:

*[…] Although I potentially support the publication of the present research, my main concern is that the effect of the mutations on the three imprinted genes is not fully understood and thus their precise role in triploid block remains unknown*.

*1) It is unclear why only one mutant allele of* SUVH7*,* PEG2 *and* PEG9 *was tested in the interploidy analysis. To avoid any misinterpretation of phenotypes, two independent mutant alleles for each gene should be tested*.

We agree with this concern and included the analysis of additional alleles for *suvh7* and *peg9*. As for *peg2* no additional alleles were available, we complemented the *peg2* mutant using a genomic construct. Data are shown in Figure 2—figure supplement 2.

*2) The authors postulate that suppression of pectin hydrolysis by* PEGs *could be the mechanism that restores triploid viability. However, this is highly speculative especially considering that the precise functions of three of the* PEGs *remain known*.

We addressed this concern by including additional data on the regulation of pectin biosynthesis and pectin degradation gene expression as well as pectin levels in diploid and triploid seeds in Figure 4. These data show that pectin degradation genes are significantly upregulated in triploid seeds and suppressed by mutations in *PEG* genes. We furthermore show that those pectin degradation genes that become upregulated in triploid seeds are expressed in the endosperm of wild-type seeds at the time of endosperm breakdown, supporting their role in pectin degradation in the endosperm. We finally show that 6 DAP triploid seeds have reduced levels of de-esterified pectins, in agreement with increased activity of pectin degradation enzymes that use de-esterified pectins as substrates.

*3) I do not see how the data support the conclusion that* PEGs *function as growth regulators and that this agrees with the predictions of the parental conflict theory*.

We have deleted this part of the manuscript.

Reviewer #2:

*In this manuscript, the authors investigated the functional roles of 15* PEGs *during* Arabidopsis *seed development. Although they did not find obvious functions of these genes in wild-type seeds, clever genetic experiments revealed that at least three* PEGs *are required for the so-called triploid block, a post-zygotic reproductive barrier that originates in the endosperm. These results substantiate their earlier findings that imprinted genes, such as* ADM*, can establish post-zygotic hybridization barriers and thus may be key speciation genes. However, they do not establish a strong connection between their genetic results and expression analyses. If the authors want to make the claim summarized in the second to last sentence in the Abstract, then more work is needed with either existing resources or new experiments.*

We have added new data supporting that *PEGs* directly or indirectly regulate pectin degradation (Figure 4). We nevertheless have removed this part from the Abstract, as this is not the major point of this manuscript, which we rather find in the fact that many *PEGs* build hybridization barriers. This finding is novel and in our view worth reporting.

*That is, while the pathways appear to converge on genes that decrease pectin degradation, which may increase cellularization and subsequent embryo viability, it is impossible to evaluate how likely this is a cause or correlation based on the presented data. Overall, I agree that this work advances the understanding of the functional roles of imprinted genes in plants, but think more could be presented to get an idea of the underlying molecular mechanism*.

We have addressed this concern; please see our comment 2 to reviewer1.

*At the very least, the authors should give more information relating to the hypothesis that suppression of pectin degradation in the triploid* PEG *mutants contributes to increased viability. This seems like an important aspect of their manuscript, but GO enrichments of commonly down-regulated genes are not necessarily strong indicators of common functions. For instance, they should mention how many genes were commonly down-regulated in the three triploid mutants (*Figure 3
*shows 70 but this was only for genes up-regulated in triploid seeds). And they should show the levels of the 18 (?) commonly down-regulated genes suggested to be involved in pectin degradation (18 based on*
Figure 3—figure supplement 1*, but perhaps some of these may be overlapping?) in wild-type, triploid and triploid* peg *mutant backgrounds*.

We have included these data in Figure 4.

*It would be interesting to see the transcript levels of these pectin degradation genes in wild-type endosperm and these should be available from the microarray datasets generated by the Harada and Goldberg labs (Belmonte (2013) PNAS)*.

We have included these data in Figure 4.

*Finally, they should stain the triploid seeds/endosperm (and controls) with a pectin dye such as Ruthenium Red to test for decreased pectin levels at multiple time points prior to cellularization. This may address whether decreased pectin is a cause or consequence of increased cellularization in the triploid mutants*.

We have included these data in Figure 4 and Figure 4—figure supplement 1.

[Editors' note: the author responses to the re-review follow.]

Reviewer #2:

*[…] The authors should perform Ruthenium Red staining on the 3n* peg *mutants (e.g. 3n* adm-2*, 3n* suv7-1*, 3n* peg2*) to check whether pectin degradation defects are rescued by mutating* PEGs*.*

As requested, we have toned down the conclusions drawn by the ruthenium red experiment.

Minor comments:

*1) Remove or reword the last sentence of the Results/Discussion. It doesn't read like science text*.

As requested, we have modified the last sentence of the Results/Discussion.

*2) Explain how seed size (*Figure 1*) was normalized*.

We included the information on seed size normalization into the Materials and methods section.

*3) Ruthenium Red staining looks less in 3n 6 DAP compared to 4 DAP. Is this representative? Also, it is interesting that there is less staining in the chalazal endosperm. Perhaps it is worth commenting on this*.

We analyzed a minimum of ten seeds of wild-type and *osd1* at 4 and 6 DAP by ruthenium red staining and the pictures shown are representative for the analyzed samples. We included this information into the Materials and methods section. Less staining in the chalazal endosperm is consistent with the fact that this region does not cellularize. We have not commented on this in the manuscript, as this was not the central point to be addressed by this experiment.

*4) In*
Figure 1—figure supplement 1*, I assume these were for seed/endosperm-specific genes. Otherwise, one would expect a strong maternal bias in all crosses. This should be noted in the text or figure legend*.

Figure 1—figure supplement 1 shows allele-specific expression data for paternally expressed imprinted genes (*PEGs*). The absence of maternal allele expression is in agreement with the fact that these genes are not expressed in maternal sporophytic tissues.

*5) Include error bars in*
Figure 1—figure supplement 3.

We included error bars in Figure 1—figure supplement 3.
